# The Regulation of Neutrophil Migration in Patients with Sepsis: The Complexity of the Molecular Mechanisms and Their Modulation in Sepsis and the Heterogeneity of Sepsis Patients

**DOI:** 10.3390/cells12071003

**Published:** 2023-03-24

**Authors:** Øystein Bruserud, Knut Anders Mosevoll, Øyvind Bruserud, Håkon Reikvam, Øystein Wendelbo

**Affiliations:** 1Leukemia Research Group, Department of Clinical Science, University of Bergen, 5021 Bergen, Norway; 2Section for Hematology, Department of Medicine, Haukeland University Hospital, 5021 Bergen, Norway; 3Section for Infectious Diseases, Department of Medicine, Haukeland University Hospital, 5021 Bergen, Norway; 4Section for Infectious Diseases, Department of Clinical Research, University of Bergen, 5021 Bergen, Norway; 5Department for Anesthesiology and Intensive Care, Haukeland University Hospital, 5021 Bergen, Norway; 6Faculty of Health, VID Specialized University, Ulriksdal 10, 5009 Bergen, Norway

**Keywords:** sepsis, neutrophils, neutrophil subsets, chemotaxis, aging, frailty, antibiotics, stem cell transplantation, metabolism

## Abstract

Sepsis is defined as life-threatening organ dysfunction caused by a dysregulated host response to infection. Common causes include gram-negative and gram-positive bacteria as well as fungi. Neutrophils are among the first cells to arrive at an infection site where they function as important effector cells of the innate immune system and as regulators of the host immune response. The regulation of neutrophil migration is therefore important both for the infection-directed host response and for the development of organ dysfunctions in sepsis. Downregulation of CXCR4/CXCL12 stimulates neutrophil migration from the bone marrow. This is followed by transmigration/extravasation across the endothelial cell barrier at the infection site; this process is directed by adhesion molecules and various chemotactic gradients created by chemotactic cytokines, lipid mediators, bacterial peptides, and peptides from damaged cells. These mechanisms of neutrophil migration are modulated by sepsis, leading to reduced neutrophil migration and even reversed migration that contributes to distant organ failure. The sepsis-induced modulation seems to differ between neutrophil subsets. Furthermore, sepsis patients should be regarded as heterogeneous because neutrophil migration will possibly be further modulated by the infecting microorganisms, antimicrobial treatment, patient age/frailty/sex, other diseases (e.g., hematological malignancies and stem cell transplantation), and the metabolic status. The present review describes molecular mechanisms involved in the regulation of neutrophil migration; how these mechanisms are altered during sepsis; and how bacteria/fungi, antimicrobial treatment, and aging/frailty/comorbidity influence the regulation of neutrophil migration.

## 1. Introduction

Sepsis is defined as life-threatening organ dysfunction caused by a dysregulated host response to infection, i.e., the host response to the infection causes organ damage/failure [1,2]. The organ functions are evaluated by the Sequential/sepsis-related Organ Failure Assessment (SOFA) score that is based on scoring respiration, coagulation, liver function, circulation, consciousness, and renal function. Finally, septic shock is observed when the circulatory dysfunctions are severe enough to substantially increase mortality; these patients can be clinically identified by the fact that they require vasopressor therapy and by their increased serum lactate levels despite adequate volume resuscitation [1,2]. Septic shock patients have a hospital mortality exceeding 40%.

Neutrophils are important effector cells of the innate immune system and constitute an important part of the first line defense against infections [3,4,5,6]. However, neutrophils derived from sepsis patients are dysfunctional (e.g., with regard to chemotaxis/migration) [3,4,5], and this dysfunction is probably important in terms of the contribution of neutrophils to the development of the sepsis-associated organ failure that directly relates to mortality and morbidity [6]. Several observations suggest that organ dysfunction/failure is not due to hypoperfusion and hypoxia alone; additional mechanisms involve endothelial and microvascular dysfunctions, immune dysregulation, and (cellular) metabolic reprogramming [6]. Our aim was to review and discuss the molecular mechanisms involved in the regulation of neutrophil migration (Section 2, Section 3, Section 4, Section 5, Section 6 and Section 7); how these mechanisms operate during sepsis (Section 8, Section 9 and Section 10) how bacteria/fungi, antimicrobial treatment, and aging/frailty/comorbidity/sex influence neutrophil migration (Section 11, Section 12, Section 13, Section 14 and Section 15); and possible therapeutic strategies that have been tried or suggested to target neutrophil functions in sepsis patients (Section 16).

## 2. The Regulation of Neutrophil Migration during Bacterial Infections: Transendothelial Migration from the Bone Marrow and Later Extravasation at the Infection Site

Sepsis is characterized by increased neutrophil release from the bone marrow followed by transendothelial migration in small vessels and extravascular chemotaxis at the infection site (see Section 3) [7,8,9,10,11]. The final step in sepsis entails reversed migration contributing to sepsis-induced organ failure. Most studies referred to in this section and in the following Section 3, Section 4, Section 5, Section 6 and Section 7 are based on studies of circulating neutrophils, and it is stated in the text when neutrophils were derived from other compartments.

### 2.1. Migration of Neutrophils from the Bone Marrow to the Circulation

Neutrophils are produced in the bone marrow, and the retention of immature neutrophils in the marrow is ensured by their expression of the CXCR4 chemokine receptor and its ligand CXCL12 [5,12,13,14,15,16]. The hematopoietic growth factor granulocyte colony-stimulating factor (G-CSF) is a downregulator of CXCR4 expression and thereby a stimulator of neutrophil migration from the bone marrow [13].

### 2.2. Extravasation of Neutrophils at the Infection Site: Molecular Mechanisms Involved in the Regulation of Endothelial Adhesion, Transendothelial Migration, and Neutrophil Polarization

Neutrophil extravasation is a multistep process [17,18,19,20]. The first step is the capture of floating neutrophils by endothelial cell surfaces via adhesion molecules that are upregulated on endothelial cells in response to proinflammatory cytokines and/or microbial molecules. These initial weak and transient interactions are mediated by the selectin family of adhesion molecules, and they allow rolling of the neutrophils on the endothelial cells. Platelet P selectin contributes to the formation of platelet–leukocyte aggregates and functions as an adhesion mechanism by bridging neutrophils and endothelium [21].

The next step is the stimulation and activation of the neutrophils by chemokines that are presented on the endothelial luminal surface. This process allows firm adhesion, and the neutrophils then crawl on the surface to finally find a site that allows transmigration [22]. The neutrophils can cross the endothelial barrier either by the parallel route between endothelial cells or by the transcellular route. Several neutrophil integrins control the firm adhesion to the endothelium including αLβ2, αMβ2, and α4β1 integrins; their ligands are intercellular adhesion molecule 1 (ICAM-1), ICAM-2, and vascular adhesion molecule 1 (VCAM-1), respectively [22].

Neutrophil polarization is thereafter necessary for directional migration; the cells develop filamentous actin polymerization at the leading edge that faces the chemotactic gradients, whereas myosin filaments assemble at the opposite end where they form the uropod [22]. Differences in the lipid composition are probably essential for the cell membrane organization of the various proteins/adhesion molecules involved in the polarization, and therefore modulation of the lipid content/structure (e.g., cholesterol depletion) will inhibit polarization/migration [22,23,24,25]. The mechanisms behind these lipid effects seem to involve the cholesterol-induced modulation of intracellular PI3-kinase/RhoA signaling that leads to uropod formation [25]. Furthermore, cholesterol is a major constituent of the plasma membrane and has important effects on the physical properties of the lipid bilayer: it regulates the lateral mobility of membrane proteins and increases the strength of membrane–cytoskeleton interactions [26]. A general effect of cholesterol depletion is therefore cell stiffening with the strengthening of the membrane–cytoskeleton interactions and increased endothelial stiffening with strengthened cytoskeleton–membrane interactions is associated with enhanced endothelial contractility and increased neutrophil migration [27]. However, other membrane lipids including phosphoinositides, phosphoinositols, lipid phosphatases, and sphingomyelins/ceramides are also involved in regulation of neutrophil polarity/chemotaxis [28,29,30].

Endothelial heparan sulfate binds the CXCL1 and CXCL2 chemokines (see Section 3.5), and these chemokines thereby become involved in the process of neutrophil crawling/diapedesis [22,31]. The chemotactic leukotriene B_4_ (see Section 3.3) and the surrounding pericytes are also regulators in these early steps of neutrophil extravasation [22,32,33]. Thus, neutrophil transmigration/extravasation is regulated by an orchestrated action of endothelial/pericyte/neutrophil adhesion molecules and chemotactic agents, and one would also expect the binding of neutrophil integrins to various extracellular matrix proteins to further modulate these initial steps of neutrophil migration.

### 2.3. Caveola Are Important for Neutrophil Transmigration and the Formation of Chemotactic Gradients

The caveolar invaginations of the neutrophil plasma membrane are important in the regulation of neutrophil migration (see Section 11.4). Caveola and caveolin-1 signaling in endothelial cells have additional indirect effects on neutrophil chemotaxis. Caveolin-1 then interacts with endothelial nitric oxide (NO) synthase resulting in the modulation of toll-like receptor (TLR)-initiated Nuclear factor κ B (NFκB) activation and transcriptional regulation, including the regulation of endothelial chemokine expression [34]. Caveolin-1-initiated signaling seems to limit inflammation through this inhibition of TLR-induced cytokine/chemokine release. Furthermore, the level of endothelial caveolin-1 (responsible for induction of membrane caveola) is important in neutrophil extravasation. Chemotaxis preferentially recruits neutrophils to a subset of venules that express high levels of ICAM-1 and low levels of caveolin-1. The level of caveolin-1 expression will then determine the balance between parallel and transcellular neutrophil extravasation [35]. In addition, extravascular chemokines can traverse the endothelial barrier when establishing the chemotactic gradients (see above), and several neutrophil–chemotactic chemokines then seem to traverse the microvasculature through caveola [35]. Thus, caveola and caveolin-1 are important both for the regulation of neutrophil transmigration and for the initiation of chemokine-dependent chemotaxis.

## 3. Regulation of Neutrophil Migration during Bacterial Infections: The Chemoattractants

Chemotaxis is defined as guided cell movement in response to chemical signals released at a distant site; the cells then move along concentration gradients formed by receptor-binding chemoattractants [7]. The four main chemoattractant are N-formylated peptide, complement factors, lipids, and chemokines. In contrast, non-directional random migration is referred to as chemokinesis [8].

Chemoattractants are classified as either end targets or intermediary targets (Figure 1) [7,9,11]. The end targets are N-formylated peptides and complement chemoattractants, and they are dominant over the intermediary chemoattractants (i.e., lipid chemoattractants and chemokines) in case of opposing gradients. These two main classes show differences in downstream receptor signaling with end target attractants mainly signaling through phospholipase A2 and/or mitogen-activated protein kinase (MAP kinase) p38, whereas intermediary attractants signal through the phosphatidylinositol 3-kinase (PI3K) pathway [10,11]. Finally, lipid chemoattractants are essential for the initial neutrophil recruitment. The first step is followed by an amplification step of cytokine/chemokine release, and the overall process is often referred to as the lipid–cytokine–chemokine cascade [9]. It can be seen from the descriptions in Section 3.1, Section 3.2, Section 3.3, Section 3.4, Section 3.5, Section 3.6 that the different chemotactic ligands/receptors form an interacting chemotactic network.

### 3.1. Formylated Peptide Receptors and Their Binding of Bacteria-Derived and Endogenous N-Formylated Peptides and Lipopeptides

Neutrophils can respond to exogenous N-formylated peptides originating from various bacteria and to endogenous peptides derived from damaged cells (e.g., mitochondrial proteins); such peptides can be recognized by three different receptors and the formylated peptide receptors (FPR) 1 and FPR2 which are both expressed by neutrophils [7,36]. FPR1 and FPR2 bind structurally diverse ligands with either agonistic or antagonistic effects [37]. Agonistic ligation of FPR1 increases neutrophil chemotaxis [38]. These formylated peptides are regarded as end target chemoattractant, i.e., when the neutrophils experience opposing chemotactic signaling, they will preferentially migrate along the formylated peptide gradients [7,36]. Intracellular signaling downstream to the FPRs involves various pathways including the p38 kinase and the Rac2 GTPase, and at high concentrations, extracellular signal-regulated kinase 1 (ERK1) and ERK2 also [39,40].

FPR1 can also bind endogenous non-formylated ligands that include peptides derived from the neutrophil mediator cathepsin G and annexin A1/annexin A4 [41,42]. The same is true for FPR2 which is more promiscuous and can bind formylated peptides as well as serum amyloid A, β-amyloid, annexin A1, humanin, and possibly lipoxin A_4_ [41,42]. FPR2 can mediate both pro- and anti-inflammatory effects depending on the ligand, and various ligands seem to differ with regard to their effects on neutrophil migration [37,41]. Possible explanations for these ligand-dependent variations are the different dimerization states of the receptor, the altered receptor conformation, and/or differences in downstream signaling depending on the ligand [41].

Lipopeptides are short endogenous or microbial-derived peptides with a fatty acid linked at the N-terminal end [37]. These molecules have both activating and inhibitory effects on G-protein-coupled receptors, and they can modulate FPR-2-initiated intracellular signaling [37]. Due to their lipid domain, the molecules pass through the cell membrane and bind to the cytosolic parts of receptor complexes. Their protein part is usually identical or similar to one of the intracellular loops of the targeted receptor, although the receptor specificity is not always complete, e.g., a CXCR4 targeting lipoprotein was also found to target FPR2 [41,42,43]. This last observation suggests that lipopeptides represent a mechanism for crosstalk between various chemotactic mechanisms [43].

### 3.2. Complement Factors

The complement factor C5 is important for neutrophil chemotaxis, and the proteolytic fragment C5a is the most potent complement chemoattractant [7,44]. C5a is formed by C5 cleavage in response to both classical, lectin, and alternative complement activation [7], and its chemotactic effect is mainly mediated through the C5a Receptor 1 (C5aR1) and its downstream mediator p38 [10,11]. Neutrophil activation through C5aR1 can also lead to the release of the chemoattractant leukotriene B_4_, i.e., C5aR1 crosstalks with lipid chemoattractants [45]. Finally, the role of complement factor C3 in inflammation is complex, but it does not seem to have any major effect on neutrophil migration [44].

### 3.3. The Interacting Lipid Chemoattractants Leukotriene B_4_ and Platelet Activating Factor

Eicosanoids derived from arachidonic acid should be regarded as the main lipid chemoattractants [7]. Leukotriene B_4_ is the strongest lipid chemoattractant for neutrophils; it is mainly released by myeloid cells and binds to the G-protein-coupled receptors leukotriene B_4_ receptor 1 (LTB4R1) and 2 (LTB4R2) [46,47]. LTB4R1 is mainly expressed by neutrophils, and receptor ligation results in neutrophil polarization followed by migration [9]. The functional effects of LTB4R2 ligation are less well characterized [7].

Another lipid chemoattractant is platelet-activating factor (PAF) which is derived from phosphatidylcholine [7,48,49]. It binds to a specific G-protein-coupled receptor on the neutrophils and can thereby activate and polarize neutrophils through intracellular signaling involving protein kinase C (PKC) [39,50]. PAF can also induce the release of leukotriene B_4_ by neutrophils [48], and there is an additional crosstalk between these two main chemotactic mediators through leukotriene B_4_ modulation of intracellular signaling downstream to PAF receptors [51]. However, animal models suggest that PAF may also inhibit neutrophil chemotaxis in certain biological contexts [50].

### 3.4. Classification of Chemokines and Chemokine Receptors

The chemokines constitute a family of low molecular weight cytokines (8–12 kDa, 70–120 amino acid residues) with chemotactic activity and, based on their chemical structure, they are classified into two main subsets: CCL and CXCL chemokines [52,53]. These two subsets bind to CCR and CXCR chemokine receptors, respectively. Several of these receptors are promiscuous with regard to ligand binding, and several chemokines are promiscuous with regard to receptor binding. The CXCL chemokines can be further subclassified into chemokines with and without a glutamate–leucine–arginine (ELR) motif before the CXC motif. CXCL chemokines with the ELR motif are CXCL1-3 and CXCL5-8; these chemokines are most important for neutrophil chemotaxis and can bind to the CXCR1/CXCR2 receptors [52,53,54]. Finally, neutrophil recruitment can also be guided by CCR1 ligands (CCL3 and CCL4), whereas CCR2 (receptor for CCL2) is most important during chronic inflammation. Finally, collagen fragments can also have chemotactic activity by binding to the CXCR2 receptor [52,53].

CCR2, CXCR1, and CXCR2 chemokines show similarities in downstream intracellular signaling. The PI3K pathway is important together with the additional activation of phospholipase C and increased intracellular calcium levels [55].

### 3.5. The Diversity of the Chemokine Receptors CXCR1/CXCR2 and Their Signaling: Biased Intracellular Signaling Is Initiated Both at the Ligand, Receptor, and Signal Initiation Levels

The chemokine system is characterized by biased signaling. This means that either (i) the relative strength of the intracellular pathway activation by two receptors with similar cellular density differs even when binding the same ligand (i.e., different ligand affinity of the receptors) or (ii) the relative activation of various pathways downstream to a certain chemokine receptor depends on the binding ligand. CXCR1/CXCR2 show biased signaling, which can be due to various molecular mechanisms [52].

*CXCR1 versus CXCR2 affinity of individual chemokines.* All CXCR2 ligands bind to this receptor with a similar high affinity, whereas the CXCR1 affinity differs between ligands with CXCL8 showing the highest affinity [56]. Thus, the balance between CXCR1/CXCR2 signaling depends on the availability of various ligand in the microenvironment.

*Chemokine receptors have two different ligand-binding sites.* The chemokines interact with CXCR1 and CXCR2 through two different binding sites, i.e., the chemokine N-terminal loop can either bind to N-terminal receptor residues (site 1) or to extracellular loop residues (site 2). Furthermore, an initial Site-1 interaction causes conformational receptor changes that are essential for binding to Site 2; differences between two chemokines in regard to the affinity for the two binding sites may therefore lead to differences in downstream signaling [57,58]. 

*Chemokine monomers and dimers differ in receptor affinity.* Chemokines exist as monomers and dimers, and some of them even exist as high-order oligomers or heterodimers [52,57,58,59]. Chemokine monomers show biological activity at the nanomolar level, but at higher concentrations, this varies. For example, the CXCL8 dimers show an altered receptor affinity with lower CXCR1 affinity but similar high CXCR2 affinity compared with the monomer [58,59,60]. Thus, the local chemokine concentration and the equilibrium/ratio between monomeric and dimeric states will influence the balance between CXCR1 and CXCR2 ligation/signaling at least for certain CXCL chemokines [59,61,62].

*Modulation of chemokine ligation by glycosaminoglycan binding.* Chemokines can bind to extracellular glycosaminoglycans, and for certain chemokines (i.e., CXCL1/5/7/8) this binding is stronger for the dimers than for the monomers [52]. Chemokine binding to glycosaminoglycans on endothelial cells is also involved in the regulation of extravasation.

*Downstream receptor signaling through G-proteins versus β-arrestin.* The intracellular signaling of chemokine receptors is initiated immediately downstream to the receptor by either G-proteins or β-arrestin [52]. G-protein-mediated signaling is rapid in onset and is followed by gradual waning, whereas β-arrestin signaling has a slower onset with sustained duration [63]. β-arrestin is additionally involved in termination of the G-protein initiated signaling through recruitment of adaptor proteins and the enhancement of endocytosis of the chemokine/receptor complex [52,59].

*Biased downstream signaling due to ligation of the same receptor by different chemokines.* Several examples illustrate that ligation of the same receptor by different chemokines leads to different downstream effects. First, CXCR2 ligation by CXCL1 leads to an influx of extracellular calcium, but this is not observed after CXCL8 ligation [64]. Second, CXCL8 bound to CXCR1 but not to CXCR2 primes neutrophils for superoxide release, and this priming effect is also observed for CXCR2 after CXCL1 and CXCL5 binding [65,66]. Third, both G-protein and β-arrestin are important for the induction of the dynamic actin remodeling that is fundamental for neutrophil migration and for the integrin activation needed for transendothelial migration, but the balance between G-protein and β-arrestin signaling varies [52,59,67,68,69]. For example, CXCR2 ligation by CXCL6 is relatively G-protein biased whereas the CXCL8 seems to be relatively β-arrestin biased with regard to cytoskeletal effects compared with other CXCR2 ligands [70]. Finally, CXCL1 monomers are more active than dimers with regard to several functional effects (e.g., calcium mobilization, chemotaxis, and β-arrestin recruitment), whereas such differences are not observed between CXCL2 monomers and dimers [60]. Finally, monomers and dimers may also differ with regard to the initiation of CXCR1/2 internalization/recycling [60,71].

*Receptor dimerization.* Dimerization may influence receptor signaling. CXCR2 forms homodimers whereas CXCR1 does not. CXCR1/CXCR2 heterodimers also exist [72,73].

To summarize, binding of the same receptor (CXCR1 and/or CXCR2) by different ligands can influence the downstream regulatory signaling of neutrophil migration.

### 3.6. Other Receptors or Mediators Involved in the Regulation of Neutrophil Migration: Molecular Mechanisms That Function as Modulators of the Main Chemotactic Pathways

Several other receptors and soluble mediators are also involved in the regulation of neutrophil chemotaxis, including:*Toll-like receptors 2 and 4 (TLR2/TLR4).* These receptors have both exogenous (e.g., microbial molecules) and endogenous agonists [74,75]. Ligation of TLR2 and TLR4 alters the neutrophil expression of CXCR1 and CXCR2 [76]. However, this effect of TLR4 may be age-dependent because it differs between newborns and adults [77]. TLR4 also increases neutrophil expression of immunostimulatory HLA-DR and the immunosuppressive T cell checkpoint molecule PD-L1 (see later Section 9.4 and Section 10.1), but these two effects may differ between neutrophil subsets [78];*TLR9.* Ligation of this intracellular receptor in neutrophils by exogenous or endogenous ligands triggers CXCR2 downregulation [79], and TLR9 ligation in other immunocompetent cells seems to have additional indirect effects on neutrophil chemotaxis [80,81,82];*NOD-like receptors.* Several cytosolic nucleotide-binding oligomerization domains (NOD) and NOD-like receptors (NLRs) have been described. NOD2 and NLRP3 are expressed by neutrophils and ligation of both receptors facilitates migration [83]. These receptors can be activated by the peptidoglycans of most bacteria [84,85];*LOX-1.* The lectin-like oxidized low-density lipoprotein receptor-1 (LOX-1) is expressed by neutrophils after TLR2/4 ligation, it activates neutrophil NFκB [86,87] but decreases CXCR2 expression in animal models [88,89];*Peroxisome proliferator-activated receptor γ (PPARγ)*. This nuclear receptor inhibits neutrophil chemotaxis in sepsis [90,91];*IL10 receptor.* The signaling inhibits neutrophil chemotaxis and cytokine release [91];*IL33 receptor.* IL33R ligation blocks TLR4-mediated CXCR2 internalization and enhances neutrophil migration in sepsis [92,93,94,95,96];*Lysophosphatidylcholines.* Lysophosphatidylcholines can modulate neutrophil migration/chemotaxis. These lipids mediate the direct effects on neutrophils by ligation of the G2A receptor, a member of the proton-sensing G-protein-coupled receptors [97]. The G2A ligation causes the recruitment of clathrin that is important for the signal transduction [97], modulates p38 signaling [98], and finally increases neutrophil expression of the αM/CD11b integrin and formylated peptide receptors [99]. In vivo animal studies have shown that these lipids can reduce neutrophil migration [100];*Sphingolipids.* Sphingolipids and sphingolipid metabolism are involved in the regulation of neutrophil migration [9,101,102,103,104,105,106]. First, lactosylceramide forms lipid raft microdomains coupled with the Src family kinase Lyn in neutrophils, and it can thereby initiate lipid raft-mediated regulation of neutrophil polarization and migration [10,104]. Second, sphingosine 1-phosphate (Sph-1-P) inhibits CXCL8- and formylated peptide-induced chemotactic migration of human neutrophils [103]. Third, glycosphingolipids seem to support slow neutrophil rolling on endothelial cells mediated by E-selectin and the transition to firm neutrophil adherence [105]. Finally, the neutral sphingomyelinase (N-SMase) is a plasma membrane enzyme that converts sphingomyelin to ceramide; it preferentially distributes towards the leading edge of neutrophils and is important for proper orientation in chemotactic gradients [30,104,106]. Thus, sphingolipids influence neutrophil migration through various molecular mechanisms.

## 4. Neutrophils Priming during Chemotaxis Modulates Neutrophil Migration

Neutrophil priming means that a primary agonist enhances the response to a second activator/event [107,108]. Priming may be transient or more durable, and the mechanisms mediating the priming effect involve various signaling pathways (e.g., p38, ERK1/2, and PI3K kinase), transcriptional regulators including NFκB, increased antiapoptotic signaling, granule mobilization, and the expression of surface molecules including the αMβ2 integrin and CD66 [109]. The ligand affinity of integrins as well as the cellular shape/deformability and polarization can be altered by priming [109]. Thus, neutrophil priming can modulate molecular mechanisms involved in neutrophil migration.

Neutrophils can be primed before (i.e., in the circulation), during, or following transmigration, and priming mechanisms that facilitate neutrophil migration involve endothelial cells, extravascular stromal cells, extracellular matrix proteins [110,111,112,113], platelets [110,111], and other immunocompetent cells [112]. Furthermore, the priming agents include the soluble mediators C3a and C5a complement components, interferon γ, CXCL8, and tumor necrosis factor (TNF) α together with pathogen-derived agents such as formylated peptides, peptidoglycan, and *Staphylococcus aureus* toxins [107,108,109,114]. Thus, neutrophils can be primed during chemotaxis, and this priming is at least partly caused by the chemotactic agents. However, there seems to be a possibility for depriming and recovery of primed neutrophils, but this process is less well characterized [109].

## 5. The Coagulation System: Regulation of Neutrophil Migration by Protease Receptors

Both activation of the coagulation system and thrombocytopenia occurs in a large number of sepsis patients [21,115,116,117]. Coagulation factors contribute to the regulation of inflammation through effects on protease-activated receptors 1-4 (PAR1-4) that can be activated by various proteases. Thrombin can activate PAR1/3/4, Granzyme A activates PAR1/2, cathepsin G activates PAR1/4, and matrix metalloprotease 1 (MMP1) activates PAR1 [118]. Several other proteases that are activated during coagulation and/or inflammation can also activate various PARs (e.g., plasmin, FVIIa, FXa, and C4a) [118]. The PARs are G-protein-coupled membrane receptors that are expressed by a wide range of cells involved in the regulation of inflammation and neutrophil migration, e.g., endothelial cells, platelets, and various leukocytes [119,120].

PARs are activated when thrombin binds to the receptor and cleaves the N-terminal domain to unmask a new N-terminus that acts as a ligand through its binding to the receptor body [120]. PAR2 agonists can then alter neutrophil expression of certain adhesion molecules (e.g., decreased L-selectin and increased α4β1 integrin), but at the same time these agonists also reduce the transendothelial migration of neutrophils [121]. This apparent discrepancy may be caused by indirect PAR effects, e.g., mediated by endothelial cells or platelets [121]. Furthermore, PAR2 agonists can also increase the neutrophil release of proinflammatory IL1β, CXCL8, and IL6 [122]. Finally, in vivo animal studies suggest that both PAR1 and PAR4 are involved in the regulation of neutrophil migration by increasing neutrophil rolling, migration, and chemotaxis [123,124].

## 6. Cytokine Release by Initially Recruited Neutrophils Increases the Further Recruitment of Neutrophils Together with a Wide Range of Immunocompetent Cells

Neutrophils can release a wide range of soluble mediators (Table 1) [125,126,127,128,129,130,131,132,133,134,135], including CXCL1/2/5/8 that bind to CXCR1 and/or CXCR2 on neutrophils; these cytokines are released through various mechanisms including exosomal release (see Section 9.9). The initial neutrophil recruitment in inflammation thereby leads to a further enhancement of the neutrophil infiltration. The other chemokines are chemoattractants for monocytes (CCL2-4), NK cells (CCL2-4 and CXCL10), mature (CCL19) and immature dendritic cells (CCL2/4/18/20), and a wide range of T cell subsets including naïve T cells (CCL18), CD8^+^ T cells (CXCL12), regulatory T cells (CCL17), Th1 cells (CXCL9-11), and Th17 cells (CCL2 and CCL20). Several neutrophil-secreted cytokines are additionally important for endothelial cell functions, angioregulation, or angiogenesis [125,126,127]. Thus, the neutrophils seem to amplify the transmigration and extravascular migration of additional neutrophils as well as a wide range of other immunocompetent cells through their cytokine release [136]. However, neutrophils also communicate and interact with their neighboring cells through several additional mechanisms, including (i) the release of extracellular vesicles/exosomes that contain various intracellular proteins and cytokines, (ii) binding to cell surface adhesion molecules and (iii) through their cytoneme, i.e., projections specialized for the exchange of mediators between cells [137].

## 7. Structural Lipids and Neutrophil Migration: The Membrane Structure, Lipid Interactions with Membrane Proteins, and the Function of Lipid Rafts

The plasma membrane is a dynamic structure assembled by lipids and proteins (see Section 2.3) [138]. The lipid composition varies between different regions of the membrane, e.g., the plasma membrane cytoplasmic leaflet usually contains more phosphatidylethanolamines and phosphatidylserines compared with the outer leaflet that is rich in sphingolipids [139]. Cholesterol deposition may also be asymmetrical, and sterols (i.e., a steroid subset including cholesterol) seem to increase membrane stiffness [140,141].

Phospholipids are important for the anchoring, orientation, and signaling of plasma membrane proteins [142], whereas gangliosides (i.e., glycosphingolipids linked to sugar chains) have effects on β1 integrins and cytoskeletal proteins that are involved in cell migration [142]. Furthermore, the modulation of fatty acid saturation is important to control the stiffness and elasticity of the cell membrane and for regulation of receptor-initiated signaling [143,144]. These lipid-associated modulations of cell signaling can be seen with relatively small changes in membrane lipid composition [145]. Finally, both the proportion of polyunsaturated fatty acids and the cholesterol level of the plasma membrane can be influenced by dietary interventions [146,147]. Although these studies were based on experimental in vitro models or animal models, the observations are possibly also relevant for neutrophil G-protein-coupled receptors.

Experimental models suggest that electrostatic interactions between the N-terminal parts of the CXCR1 receptor and the lipid parts of the cell membrane are important for the three-dimensional folding of the receptor chain [148,149,150,151]. These interactions increase the affinity of CXCR1 for CXCL8 [148]. Furthermore, after CXCR1 ligation by CXCL8, the receptor relocalizes to lipid rafts that facilitate its G-protein-mediated signaling [152]. The leukotriene B_4_ receptor BLT1 is also localized to lipid rafts in neutrophils, and cholesterol deprivation disrupts the lipid rafts and thereby reduces downstream BLT1 signaling [153,154]. In contrast, the FPR-1 receptor is localized outside the lipid rafts and its downstream signaling is therefore not altered by lipid raft disruption [153].

## 8. Modulation of Neutrophil Migration in Patients with Sepsis: Reverse Migration Due to Inflammation-Induced Modulation of Normal Chemotactic Mechanisms

Most neutrophil studies in sepsis patients (Section 8, Section 9 and Section 10) are based on circulating neutrophils; it is stated in the text when the cells were derived from other compartments.

The local clearance of neutrophils at inflammatory sites is caused by local necrosis or the induction of apoptosis with subsequent phagocytosis by macrophages, but neutrophils can additionally migrate back to the circulation [155]. Animal models have shown that up to 70% of recruited neutrophils can return to the circulation from inflammatory sites [156,157]. Reverse migrated neutrophils are characterized by a ICAM-1^high^CXCR1^low^ phenotype, and this is different from the majority of circulating ICAM-1^low^CXCR1^high^ and tissue-resident ICAM-1^high^CXCR1^high^ neutrophils [158,159]. The multistep process of reversed neutrophil migration is regarded as a programmed process characterized by defined molecular events [158,159,160,161,162,163,164,165,166,167,168,169,170]:*Modulation of chemotactic responses.* Neutrophils from sepsis patients show attenuated chemotaxis towards formylated peptides, leukotriene B4, and CXCL8 [170]. Furthermore, neutrophil C5aR expression reaches a peak early during sepsis and thereafter gradually declines [170];*Altered microvascular permeability modulates chemokine gradients.* The endothelial junctions are altered during inflammation, the microvascular permeability is thereby increased and the leakage of chemokines (e.g., CXCL1) from the circulation to the perivascular environment alters the chemokine gradients and contributes to reversed neutrophil migration [160,161,162]. Animal models suggest that CXCL8 and CXCR2 are also important for reversed neutrophil migration [164]; CXCL8 is a chemoattractant at lower concentrations but it can function as a chemorepellent at higher levels and thereby facilitate reversed migration [165];*A possible role of altered receptor trafficking.* Chemokine receptors can be inactivated by internalization and degradation, and end-target chemoattractants use this mechanism to desensitize neutrophils to intermediary chemoattractants [168]. Receptor levels can be quickly altered by the modulation of receptor internalization/degradation compared with mechanisms that require altered gene transcription. Animal models suggest that altered CXCR1/CXCR2 balance due to continued CXCR2 signaling together with reduced CXCR1 levels/signaling facilitates reverse neutrophil migration [153,158,159,168];*CXCR4/CXCL12 signaling.* Disruption of CXCR4/CXCL12 signaling would facilitate reversed neutrophil migration, but animal models suggest that CXCR4 is upregulated when reversed migrated neutrophils migrate to new peripheral organs [166]. These neutrophils show increased expression of the early apoptotic marker Annexin V, and the CXCR4 upregulation may facilitate their later migration back to the bone marrow [166]. Thus, at least some reverse migrated neutrophils possibly migrate via other peripheral organs to the bone marrow where they undergo apoptosis;*Local release of neutrophil elastase alters endothelial cell junctions.* Animal studies suggest that neutrophils can exhibit extravascular-to-luminal migration through endothelial cell junctions, and this process depends on the reduced expression and/or function of the endothelial junctional adhesion molecule-C (JAM-C) [167]. Neutrophil elastase cleaves JAM-C and thereby promotes neutrophil-reversed migration [167]. Furthermore, the local release of neutrophil elastase is promoted by leukotriene B_4_; this chemotactic signal thereby seems to promote reverse transendothelial migration of neutrophils during the late steps of inflammation [167]. Finally, neutrophil elastase seems to be presented to JAM-C by the neutrophil αMβ2 integrins (CD11b/CD18), but ICAM-1 (expressed at high levels by reversed migrating neutrophils) also seems to be important for the cleavage of JAM-C [167];*Arachidonic acid metabolism.* Leukotriene B_4_ is an early proinflammatory derivative of arachidonic acid that seems to facilitate the later resolution of inflammation through the release of neutrophil elastase (see above). Furthermore, neutrophils seem to shift to the production of the alternative anti-inflammatory arachidonic acid derivative Lipoxin A_4_ when they are localized in an inflammatory environment [168], and Lipoxin A_4_ enhances reversed migration [169];*Proteases.* The levels of proteases other than neutrophil elastase may also be involved in the regulation of reversed migration, e.g., Cathepsin C reduction reduces reversed neutrophil migration [166];*Hypoxia-inducible factor 1 α (HIF-1α)*. The transcription factor HIF-1α is expressed by activated neutrophils; it is targeted for degradation by oxygen-dependent propyl oxygenase but this enzyme is suppressed in response to bacterial infection and HIF-1α will therefore accumulate [168]. Activated HIF-1α promotes neutrophil survival and reduces reversed migration [166,168], but it is not known whether or how HIF-1α is involved in the regulation of reverse neutrophil migration;*Macrophages.* Concomitant macrophage infiltration can also facilitate reversed neutrophil migration [155].

To conclude, the molecular mechanisms responsible for reversed neutrophil migration are complex and involve the modulation of several mechanisms that are important for the early steps of proinflammatory extravasation/tissue migration, i.e., arachidonic acid metabolism, CXCR1/2 signaling, neutrophil/endothelial adhesion, and microvessel permeability. Whether modulation of neutrophil retention signaling (e.g., CXCR4/CXCL12, HIF-1α, and semaphorins) also contributes needs to be further investigated [165,166,168,170].

## 9. Modulation of Neutrophil Migration and Communication in Patients with Sepsis; Several Cellular Mechanisms Are Altered by the Infection

### 9.1. Increased Bone Marrow Release of Neutrophils: Effects of Sepsis on G-CSF and CXCL12

Inflammation is often associated with increased levels of circulating neutrophils due to increased bone marrow release [171,172]. The growth factor G-CSF can be released by neutrophils together with a wide range of proinflammatory cytokines (Table 1); it is upregulated in sepsis as a part of the acute phase reaction and promotes the generation and release of mature and immature neutrophils [173,174]. Furthermore, under homeostatic conditions, CXCL12 is highly expressed in bone marrow stromal cells, and CXCL12/CXCR4 serves as a main mechanism for bone marrow retention of neutrophils [175]. Infectious stress causes rapid downregulation of CXCL12 mRNA and protein levels in the bone marrow, and neutrophil release is thereby facilitated [171,172].

### 9.2. Effects of Sepsis on Neutrophil Rigidity and F-Actin Accumulation: Increased Rigidity as a Possible Mechanism That Increases the Risk of Organ Failure

Proinflammatory mediators as well as bacterial products cause increased neutrophil rigidity and priming of the neutrophils at the same time [176,177]. The increased rigidity is associated with F-actin accumulation; it can be induced by proinflammatory TNFα and this TNFα effect seems to be mediated by the transcription factor PPARγ [90,178]. There is also reduced endothelial rolling of neutrophils during sepsis, and increased rigidity/priming combined with decreased rolling probably leads to neutrophil sequestering in the microcirculation as a part of the pathogenesis of organ dysfunction/failure [177,179].

### 9.3. Fatty Acids, Cell Membrane Structure, and Neutrophil Migration in Sepsis: Dietary Polyunsaturated Fatty Acids Decrease the Mortality in Experimental Sepsis

Recent studies investigated the effect of dietary polyunsaturated fatty acids in a murine model of *Staphylococcus aureus* sepsis [180,181,182,183]. Increased dietary polyunsaturated fatty acids then increased survival, decreased the bacterial load, and increased neutrophil infiltration at the inflammatory site. The cell membrane should be regarded as a dynamic regulator of neutrophil migration [144,145,146,184,185]. A possible explanation for this effect of dietary polyunsaturated fatty acids could therefore be modulation of neutrophil membranes with altered chemotactic signaling, increased neutrophil migration/infiltration, and thereby decreased mortality in sepsis. Alternatively, dietary modulation may alter the membrane stiffness and thus have a general effect on neutrophil migration that reduces the risk of multiorgan failure.

### 9.4. Effects of Sepsis on Neutrophil–Endothelial Interactions: The Importance of Adhesion Molecules, Integrins, and Heparinase

Studies in animal models have demonstrated that endothelial ICAM-1 expression is upregulated during sepsis, and there is also a transient increase in VCAM-1 [186,187,188,189,190]. Sepsis is also associated with increased degradation of the glycocalyx, and a TNFα-induced increase in endothelial expression of heparinase is important for this degradation [191]. A possible effect of this degradation is exposure of adhesion molecules on the endothelial cells leading to increased neutrophil adhesion.

Neutrophils upregulate β1 and β2 integrins in response to proinflammatory mediators; these adhesion molecules interact with endothelial ICAM-1/VCAM-1 and studies in animal models suggest that this upregulation enables firm adhesion to endothelial cells and thereby inhibition of the later extravascular neutrophil migration [95,96,192]. Animal models suggest that the combined upregulation of integrins in circulating neutrophils and endothelial adhesion molecule is seen especially for gram-positive bacteria, whereas for other bacteria the integrin upregulation is seen only for interstitial neutrophils [193]. Animal models also suggest that circulating neutrophils upregulate β1 integrins during sepsis, and this upregulation may influence neutrophil crawling and adhesion to endothelial cells [194,195].

Proteases can modulate endothelial cell regulation of inflammation through the ligation of protease-activated receptors (see Section 5), but further studies are needed to clarify whether these ligand/receptor interactions are involved in the pathogenesis of sepsis-associated organ failure.

### 9.5. Selectins and Selectin Receptors in Sepsis

Neutrophil L-selectin and its ligands (CD44, PSGL1 and ESEL-1) are important for neutrophil adhesion to endothelial cells [196,197], but the possible modulation of selectins/selectin ligands during sepsis has not been characterized in detail [197].

### 9.6. Soluble Adhesion Molecules as Modulators of Endothelial–Neutrophil Interactions in Sepsis?

Several adhesion molecules exist in biologically active soluble forms, and this is true both for selectins and ICAM-1 [198]. Membrane-bound forms of these adhesion molecules are expressed by endothelial cells and/or neutrophils and are important for extravasation and migration (see Section 2.2). The systemic levels of these soluble adhesion molecules can be altered in patients with bacterial infections/sepsis [198,199]. The serum levels of L-, P-, and E-selectins as well as ICAM-1 are then increased in immunocompetent patients. In contrast, patients with chemotherapy-induced neutropenia show only increased serum levels of L-selectin and ICAM-1 whereas P- and E-selectin levels are decreased during bacterial infections [198,199]. The possible influence of these infection-induced alterations on in vivo neutrophil migration has not been investigated, but the biologically active soluble forms will possibly compete with the corresponding membrane-expressed forms and thereby influence neutrophil migration [198].

Several other cell membrane molecules can also be released by neutrophils during sepsis [200]. First, CD14 is a part of the TLR4 receptor complex, and it is upregulated and released from the lipopolysaccharide-binding protein in sepsis. Second, the triggering receptor expressed on the myeloid cells 1 (TREM1) is also upregulated and released by neutrophils after exposure to bacteria. Third, azurocidin is released when neutrophils bind to selectins, and this protein can alter the endothelial cytoskeleton and thereby enhance neutrophil migration through the endothelial layer. However, additional studies are needed to clarify whether these soluble forms will influence neutrophil migration and/or can be used as a biomarker of sepsis in routine clinical practice.

The serum levels of soluble VCAM-1 and ICAM-1 correlate with organ dysfunction in sepsis [186,201,202,203], and these clinical observations support the hypothesis that soluble adhesion molecules are involved in the pathogenesis of sepsis.

### 9.7. Multifactorial Modulation of Neutrophil Chemotaxis in Sepsis

The regulation of neutrophil chemotaxis shows complex alteration in patients with sepsis [204,205,206,207,208,209,210,211,212,213,214,215,216,217,218,219,220,221,222,223,224,225,226,227,228,229,230,231,232,233,234,235,236,237,238,239,240,241,242,243,244,245,246,247,248,249,250,251], and Table 2 summarizes how various neutrophil receptors/mediators are involved in the sepsis-associated modulation of neutrophils.

*General inhibition of neutrophil chemotaxis and associations with prognosis.* Studies in animal models have described inadequate neutrophil migration towards infection foci during severe sepsis even though high chemokine levels are seen at the infection sites [74,204]. These abnormalities are probably caused by reduced responsiveness for all four main chemotactic mechanisms (see Figure 1). The migratory responses to formylated peptides, LTB_4_ and CXCL8 are reduced, and neutrophil C5aR expression reaches an early peak during sepsis before a gradual decline [154]. Furthermore, patients with the largest reduction of neutrophil migration towards formylated peptides or LTB_4_ have decreased survival [205]. Decreased neutrophil migration towards CXCL8 is seen especially in patients with severe sepsis; this chemotactic CXCL8 effect seems to be a part of a more complex dysregulated phenotype also including defects in NETosis and delayed apoptosis [206]. Finally, animal studies suggest that C5aR dysfunction alters neutrophil migration and thereby contributes to the development of organ failure [207,208,209].

*The role of G-protein-coupled receptor (GPCR) kinases.* Repeated exposure to agonists will lead to the internalization of many G-protein-coupled receptors, and this process seems to depend on G-protein-coupled receptor kinases (GPCR kinases) [210]. These kinases phosphorylate intracellular parts of activated receptors and facilitate the recruitment of arrestins that decouple the receptor from the G-protein and thereby lead to receptor internalization [211]:TLR2/4/9 ligation induces the upregulation of GPKR kinase 2 and thereby internalization of CXCR2 [74,79,93,212,213]. In contrast, IL33 prevents TLR-induced upregulation of GPCR kinase 2 and can thereby stimulate neutrophil chemotaxis [93,214];The TLR induction of CXCR2 internalization seems to depend on TNFα [235,236,237,238,239,240,241]. This TNFα effect is possibly mediated by an autocrine loop because TNFα can be released by neutrophils (Table 1), but TNFα release by neighboring immunocompetent cells (see Section 6) may also contribute to local effects and to the increased systemic TNFα levels in sepsis patients [215];Both TLR and TNF-dependent pathways upregulate NO synthase that in turn induces GPCR kinase 2 and finally reduces neutrophil expression of CXCR2 [216]. Taken together these observations suggest a signaling cascade involving TLR→TNFα→NO synthase→GPCR kinase 2→CXCR2 downregulation;Finally, PI3Kγ-initiated signaling can also cause GPCR kinase induction and thereby reduced CXCR2 expression [217,218].

Thus, there seems to be three different intracellular signaling events/pathways (i.e., TLR, TNFα, and PI3Kγ) that contribute to the internalization and thereby the reduction of CXCR2 cell surface expression in neutrophils in sepsis.

*CXCR1 versus CXCR2 expression.* As explained above, CXCR2 is internalized in circulating neutrophils during sepsis [219,220,221]. Decreased neutrophil CXCR2 expression in sepsis patients correlates with the Apache II score, a parameter for disease severity [219]. In contrast, neutrophil CXCR1 levels are not altered in sepsis patients [219].

*The role of the CCR2 chemokine receptor.* CCR2 is induced on the neutrophil surface during sepsis in a TLR2/4 dependent manner; this promiscuous chemokine receptor is probably not involved in neutrophil recruitment to the infectious site [223], but it seems to be involved in neutrophil recruitment to vital organs and the pathogenesis of organ failure [223,224]. Finally, non-surviving septic shock patients have higher neutrophil CCR2 expression than survivors [224].

*The role of Toll-like receptors.* Neutrophil expression of TLR2 and TLR4 seems to be downregulated in patients with septic shock, and gene expression studies show an additional downregulation of several inflammatory response gene clusters, including TLR-dependent genes [225,226]. However, despite this TLR downregulation, these receptors mediate important inhibitory effects on CXCR3 expression (see above).

*Platelet-activating factor (PAF).* The PAF receptor activates MAP kinases and induces the production of proinflammatory cytokines including CXCL8 [227,228]. However, despite the increased cytokine releases the final effect of receptor ligation is reduced neutrophil migration, and receptor blocking increases survival in animal sepsis models [227,228].

*Hydrogensulfide-synthesis and its influence on endothelial adhesion.* The activity of the hydrogen sulfide-synthesizing enzyme cystathionine g-lyase is increased in sepsis; its increased activity seems to be associated with increased neutrophil adhesion to endothelium, increased migration/chemotaxis, and maintained/increased CXCR2 expression in murine sepsis [229,230].

*The possible importance of LOX-1, IL17, and Peroxisome proliferator-activated receptor γ (PPARγ).* Both LOX-1 and the acute phase protein α-1 acid protein contribute to the failure of neutrophil migration [89,231,232]; this is possibly mediated by the LOX-1 receptor recognition of several inflammatory products including CRP, apoptotic cells, various bacterial products, and activated platelets [89,233]. Finally, IL17 seems to facilitate the recruitment of neutrophils to the infection site [234], whereas the results from the studies of PPAR effects on neutrophil migration in sepsis are conflicting [204].

### 9.8. Reversed Neutrophil Migration in Patients with Sepsis: Is It a Sign of Immunological Imbalance and a Mechanism in the Pathogenesis of Sepsis-Associated Organ Dysfunction/Failure?

It is well-accepted that immunological dysregulation is essential in the development and progression of sepsis [204]. Neutrophils then have effector mechanisms that are essential for the removal of the infecting agent, including degranulation, phagocytosis, and the formation of extracellular traps, but they are also important for local immunoregulation at the infectious site and in the pathogenesis of distant organ dysfunctions [204].

Neutrophils are among the first immunocompetent cells that are recruited to an infectious site; they become important through several mechanisms that amplify inflammation during the progression of sepsis. Reversely migrating neutrophils still seem to have a proinflammatory phenotype characterized by their increased capacity of superoxide release, high ICAM-1 membrane expression, likely increased levels of inducible nitric oxide synthase, increased capacity of extracellular trap formation, and increased lifespan due to antiapoptotic signaling [252,253,254,255,256]. Experimental studies have also shown an association between the level of reverse-migrated neutrophils in the circulation and organ failure [253,257,258]. Taken together these observations are consistent with the hypothesis that reversed migration contributes to the systemic dissemination of the inflammatory response and to organ failure distant to the infection site. Experimental animal models of sepsis also suggest that LTB_4_ promotes reverse neutrophil migration in sepsis and thereby contributes to the pathogenesis of organ failure [257,258]. Thus, even though available data on reversed neutrophil migration in human sepsis are limited, experimental evidence suggests that reversed migration is involved in the pathogenesis of sepsis-associated organ failure.

### 9.9. Exosomes in Sepsis: A Possible Mechanism for Neutrophil Communication with Neighboring Cells and for Their Influence on Distant Organs

Exosomes are extracellular 40–150 nm endosome-derived vesicles; they carry proteins and nucleotides/noncoding RNA as well as metabolites/lipids that can be transferred between cells [259]. Animal models have shown that neutrophil-derived exosomes are increased in the blood during sepsis, and these exosomes carry high levels of proinflammatory mediators, including cytokines and DAMP molecules. The exosomes probably represent an important form of communication between neutrophils and neighboring as well as distant cells/organs, and they possibly have a role in the development of distant organ dysfunctions. Studies in hepatocytes have shown that CXCR1 and CXCR2 regulate exosome release [260], but it is not known whether similar effects are seen in neutrophils during chemotaxis. The effect of microparticles on neutrophil migration depends on the cell surface molecules of the particles. Some microparticles seem to enhance neutrophil chemotaxis to CXCL8, and this is mediated by L-selectin and P-selectin glycoprotein ligand-1 present on the surface of these particles [261]. In contrast, particles may also mediate anti-inflammatory effects and inhibit neutrophil chemotaxis; this effect is associated with the surface expression of annexin-1 [262].

## 10. Several Molecular Markers of Neutrophil Heterogeneity Are Involved in the Regulation of Neutrophil Migration: The Possible Importance of Various Subsets in Sepsis

The modulation of neutrophil migration occurs in a complex context of a more general sepsis-induced immunosuppression [263,264]. The complexity is further increased by the identification of several neutrophil subsets that seem to differ with regard to regulation of migration [265,266,267,268,269,270,271,272,273,274,275,276,277,278,279,280,281,282,283,284,285,286,287,288,289,290,291,292,293,294,295,296,297,298,299,300,301,302,303,304,305,306].

### 10.1. Neutrophil Heterogeneity

The circulating neutrophils were previously regarded as fully differentiated phagocytic defender cells, but they are now regarded as a heterogeneous population including different effector and immunoregulatory subsets [270,284]. Several of the molecular markers used to identify various neutrophil subsets are important regulators of fundamental neutrophil functions, including neutrophil migration (Table 3). The following neutrophil subsets were identified:Mature or classic neutrophils have the phenotype CD16^+^ (Fc receptor), CD177^+^ (cell surface glycoprotein/CD32 binding), CTL-2^+^ (von Willebrand’s factor receptor), CD11b/CD18^+^, and CD11a/CD18^+^ (two integrins) [281]. An alternative strategy to identify neutrophil subsets is based on differences in cell density [284,285,286], and the low-density cells can then be further subclassified based on their expression of CD11b, CD16, and CD86 [284];Animal studies suggest that aged neutrophils have a different phenotype [287], but the survival time seems to differ between neutrophil subsets and cells with the HLA-DR^+^CD49^+^CD80^+^ phenotype have an extended survival of up to 72 h [288];CD177 is a cell surface glycoprotein; most humans have both CD177^negative^ and CD177^positive^ neutrophils [289]. One study investigated 535 healthy individuals: 65% of them had >60% CD177^+^ circulating neutrophils, 25% had intermediate levels (20–60% CD177^+^ neutrophils), and a small minority of 14 patients had only CD177^-^ neutrophils. Proteinase 3 is a neutrophil intracellular protease; the fraction of circulating protease 3 positive neutrophils in healthy individuals varies between 0 and 100% [282] and there is a cellular covariation between CD177 and protease 3 [283];The minority of mature CD66b^+^CD10^+^ neutrophils differ in their regulation of T cell activation compared with CD66b^+^CD10^-^ neutrophils [281,290], a subset of neutrophils seem to have a specialized B cell helper function [291], and neutrophil subsets differ even in their capacity to support cancer cell proliferation [284].

Thus, several neutrophil subsets were identified, and animal studies suggest that the balance between neutrophil subsets can be important for the capacity to resist certain bacterial infections [281,292]. Furthermore, studies in human inflammatory diseases suggest that the neutrophil heterogeneity is even more extensive, including CD63^+^ or PD-L1^+^ subsets identified in patients with cystic fibrosis [293,294], CD64^+^ neutrophils as a marker of sepsis (see below) [270], subsets expressing the receptor activator of nuclear factor κB ligand (RANKL) identified in chronic obstructive pulmonary disease [295], CD49d^+^ neutrophils detected in the nasal fluid during viral infections [296], and HLA-DR^+^ subsets in patients with Leishmaniosis [297]. It is difficult to judge whether these subsets represent true differentiation or whether they are markers of activation, and it is not known whether these subsets can be detected in patients with sepsis.

### 10.2. Neutrophil Expression of the Fcγ Receptor CD64 in Patients with Sepsis: Increased Levels That Associate with Prognosis

The CD64 FcγR1 is expressed only at low levels on resting neutrophils; this expression is increased during sepsis, and the level of circulating CD64^+^ neutrophils is regarded as a possible diagnostic or prognostic biomarker in sepsis [272,273]. CD64 expression is increased early during sepsis, and the levels are generally higher for patients with septic shock. Furthermore, the CD64 level correlates with severity scores such as APACHE II and total SOFA scores and predicts survival/mortality for intensive-care sepsis patients. A recent study of sepsis patients showed that immature CD64^+^ neutrophils could be further subclassified; two new neutrophil subsets were then identified based on the expression of CD123 (the IL3 receptor) and PD-L1 [298]. Thus, CD64^+^ neutrophils are heterogeneous, but it is not known whether the diagnostic/prognostic impact of circulating CD64^+^ neutrophil levels is caused by all or only a subset of the CD64^+^ neutrophils.

### 10.3. Neutrophils from Patients with Sepsis Show Increased CD177 (a Cell-Surface Glycoprotein) and Decreased CD10 Expression, and the Levels of the Two Markers Show Inverse Correlation

A study of neutrophils derived from 45 patients with septic shock identified 364 upregulated and 328 downregulated genes. CD177 mRNA had the highest fold change alteration when patients were compared with healthy individuals, and this difference was also confirmed at the protein level [299]. However, the sepsis patients were also characterized by a constant CD177 negative neutrophil subset, CD177 was coexpressed with protease 3, and circulating neutrophils in addition showed decreased CD10 levels with a significant inverse correlation between CD177 and CD10 protein levels. The CD10 metalloprotease is a negative regulator of chemotaxis [279], and the increased CD177/decreased CD10 combination will thus facilitate the extravasation/migration of this subset (Table 4).

### 10.4. The Fraction of Olfactomedin 4 (OLFM4) Positive Neutrophils Is Increased in Patients with Sepsis, and High Levels Are Associated with Adverse Prognosis Even in Adjusted Analyses

OLFM4 is a neutrophil-specific granule protein secreted through exosomes [278]. The percentage of OLFM4-expressing neutrophils in healthy individuals varies between 8 and 57% (Table 4), and the levels of positive cells increase in sepsis. However, the levels can also be increased especially in other bacterial infections, but also in viral and parasitic infections, sterile inflammatory disorders, and in cancer patients [274,278,300].

A recent study described the increased mortality of sepsis patients with high levels of circulating OLFM4^+^ neutrophils and the same was true for non-septic patients with infections and Systemic Inflammatory Response Syndrome (SIRS). In total, 56% of sepsis patients with elevated OLFM4^+^ neutrophils died compared to 18% of patients with OLFM4^+^ neutrophils below 37.6% [301]. These associations between the OLFM4^+^ neutrophil subset and mortality were significant when adjusting for age, sex, absolute neutrophil count, comorbidities, and the total SOFA score. Furthermore, similar observations were made in a prospective pediatric study including patients with septic shock; patients with at least two organ failures at day 7 of septic shock or dying before day 28 had a higher percentage of OLFM4^+^ neutrophils compared to patients with less severe disease [302]. These associations were also significant after adjusted analyses. Thus, high levels in the circulation of the OLFM4^+^ neutrophil subset are associated with severe/multiple organ failures, and this cell subset may therefore have a role in the pathogenesis of organ failure [302,303,304].

OLFM4 can be released by neutrophils, and the serum level of OLFM4 is increased in sepsis patients, especially patients with severe disease [303]. Finally, OLFM4 inhibits cathepsin C which activates cathepsin G [305], and cathepsin G is a regulator of neutrophil chemotaxis [195,276,277,278,306]. Thus, the modulation of neutrophil migration may contribute to the adverse prognostic impact of the ILFM4^+^ neutrophil subset and/or soluble OLFM4.

### 10.5. Myeloid-Derived Suppressor Cells in Sepsis; Further Neutrophil Development to an Immunosuppressive Neutrophil Subset Associated with Adverse Prognosis in Sepsis Patients

The two main myeloid-derived suppressor cell (MDSC) subsets are monocytic and granulocyte (G)-MDSC; the latter has the phenotype CD33^+^CD11b^+^CD14^+^HLA-DR^low^ [307]. Thus, G-MDSCs represent a subset or a further neutrophil differentiation/development. Several neutrophil-secreted cytokines can function as expanders of MDSCs, and the TLR-target NFκB can function as an MDSC activator [307].

There is an increased expression of the PD-1 ligand by MDSCs in sepsis patients, and ligation of the PD-1 checkpoint on T cells leads to T cell apoptosis that can contribute to the observed lymphopenia during sepsis [307]. Furthermore, high G-MDSC levels are associated with adverse prognosis in sepsis patients, whereas a similar association is not observed for monocytic MDSCs [306,308,309,310]. Furthermore, typical mature G-MDSCs as well as a CD14^low^ mature G-MDSC subset show increased levels especially in gram-positive sepsis [311]. Finally, low CD10 (CD10^dim^) and/or CD16 (CD16^dim^) expression by G-MDSC were associated with an adverse prognosis in sepsis [312]. Thus, G-MDSCs subsets are included among neutrophils and increased MDSCs seem to contribute to the immunocompromised state and an adverse prognosis in sepsis.

## 11. Important Bacterial Products That Interfere with Neutrophil Functions

Sepsis can be caused by a wide range of bacteria; gram-positive infections have increased over time and are now almost as common as gram-negative infections [313,314]. This increase is probably due to the increased use of invasive procedures. Various *Staphylococcus* species are the most common among gram-positive infections (30–35% of cases), although *Enterococcus* infections (10%) are also common. A wide range of gram-negative bacteria can cause sepsis (55–60%), whereas fungal infections constitute a smaller subset (15%) with Candida species being the most common. The most common sites of infection are respiratory (40% of cases), genitourinal (15–20%), and abdominal (5–10%), but 6–8% of patients have an unknown site [84,85,314,315,316]. In this section, we will describe how common sepsis-causing bacteria interact with neutrophils and thereby modulate their migration [84,85,214,243,269,317]. Most neutrophil studies referred to in this section and the following Section 12, Section 13, Section 14, Section 15 and Section 16 are based on circulating neutrophils; it is stated in the text when cells were derived from other compartments.

### 11.1. Modulation of Neutrophil Migration by Staphylococcus aureus

The effects of *Staphylococcus aureus* on neutrophils, including neutrophil migration, have been extensively studied, and several bacterial molecules can modulate neutrophil migration [107,318]:The chemotaxis inhibitory protein of *Staphylococcus aureus* (CHIP) can bind and thereby block the extracellular parts of the chemotactic C5a and formylated peptide receptors but without initiation of intracellular signaling [319,320,321,322,323,324];Agonistic ligation of the neutrophil formyl peptide receptor-like-1 stimulates neutrophil migration [38]. *Staphylococcus aureus* releases additional proteins that bind to and block this receptor and thereby inhibit neutrophil chemotaxis [317,324,325]. Some of these antagonists also block Fcγ receptors [107,318];The cysteine protease Staphopain A cleaves the N-terminal part of the CXCR2 receptor and thereby inhibits its binding of chemokine ligands [326];Selectins are important for the initial steps of neutrophil extravasation, and neutrophils express the P-selectin glycoprotein ligand 1 [22,107,318]. *Staphylococcus aureus* secretes two proteins that bind to this selectin ligand and thereby inhibit neutrophil rolling on endothelial cells [327,328]. One of these proteins can also bind to the N-terminal end of other G-protein-coupled receptors and thereby inhibit their binding with other ligands; this mechanism inhibits effects of complement factors and CXCL chemokines on neutrophil migration [329];TLR2 can recognize several staphylococcal lipoproteins that have a TLR2 antagonizing effect [330,331];Staphylococcal superantigen-like proteins 1 and 5 are broad protease inhibitors; they inhibit the neutrophil proteases MMP8 and MMP9 and thereby the proteolytic cleavage and potentiation of CXCL8 [332].

These observations clearly illustrate that bacterial molecules can modulate/inhibit neutrophil migration through various molecular mechanisms. These mechanisms may also contribute to the increased levels of certain MDSC subsets especially in gram-positive infections [311].

### 11.2. Streptococcal Modulation of Neutrophil Migration

Several cell wall components derived from *Streptococcus pneumoniae*, including lipoteichoic acid and other lipoproteins, are recognized by TLR2, and downregulation of chemokine receptors would therefore be expected [242]. Furthermore, TLR4 can bind the exotoxin pneumolysin, and pneumococcal DNA can bind to TLR9 [242]. One would expect ligation of these TLRs to cause downregulation of chemokine receptors and thereby inhibition of neutrophil migration (see Section 9.7).

### 11.3. Gram-Negative Bacteria; TLR4 Binding and Binding of Microorganisms to Lipid Rafts

Lipopolysaccharide is a ligand for TLR4 [243,333], and gram-negative bacteria would therefore be expected to inhibit neutrophil migration through TLR4-mediated downregulation of chemokine receptors (see Section 9.7).

Lipid rafts are membrane domains rich in sphingolipids/cholesterol, and they contain glycosylphosphatidylinositol-anchored proteins as well as other signaling molecules also anchored via lipids (see Section 7) [334]. The lipid rafts are important for chemotactic signaling in neutrophils, but certain components can also function as receptors for pathogen-associated molecular patterns [334,335]. This is true for lactocylceramides that are able to bind directly to pathogen components including molecules derived from mycobacteria and candida species, such as *Pneumocystis jeroveci* beta-glycan and *Escherichia coli* proteins [335]. Other glycosphingolipids are also able to bind microbiological molecules including *Escherichia coli*, *Pseudomonas aeruginosa,* and shigella enterotoxins. Thus, the glycosphingolipids in lipid rafts are able to function as pathogen-associated molecular pattern receptors, and at least some of these (e.g., soluble beta-glucan from *Candida albicans*) can induce neutrophil migration [334,335].

Caveola are plasma membrane invaginations with a diameter of 60–80 nm; they are a subset of lipid rafts enriched in cholesterol and sphingolipids (see Section 2.3 and Section 7) [34]. Caveolin is a major constituent of the caveola; it exists in the isoforms caveolin-1-3 and is involved in cholesterol transport [336,337]. However, several G-protein-coupled receptors are sequestered within the caveola through interactions with caveolin-1, a protein that is expressed by many different cells including neutrophils [338]. Caveolins can also bind various microorganisms [34]. Furthermore, caveolin-1-deficient mice show increased mortality during infections with *Salmonella* and *Pseudomonas aeruginosa,* but at the same time, these mice have increased neutrophil infiltration at the infection site [34]. Thus, caveola are involved both in microbial binding as well as in G-protein-coupled receptor signaling, and experimental models suggest that caveolin is important for neutrophil migration [338]. Future studies have to clarify whether caveola-mediated effects on neutrophil migration are caused by binding to neutrophil caveola or whether indirect effects mediated by binding to endothelial caveola, for example, also contribute.

### 11.4. Candida Species and Neutrophil Migration

Candida species and especially *Candida albicans* show several molecular interactions that can modulate neutrophil migration and the proinflammatory neutrophil phenotype. First, various chemotactic formyl peptide receptors bind fungal molecules [339]. Second, the αMβ2 integrin has two distinct ligand-binding sites that cause different cellular responses; one of these sites is highly promiscuous and can bind both endogenous and microbial ligands [340]. Dual ligation of αMβ2 by the extracellular matrix protein fibronectin and candida β-glucan will then enhance neutrophil chemotaxis, and at the same time, there is a shift in fibronectin binding from α5β1 to α3β1 integrins. Thus, these effects of candida on neutrophils are caused by other molecular mechanisms aside from the bacterial effects.

Recent studies have described that the β-1,6-long glucosyl side-chain-branched beta-glucan derived from *Candida albicans* causes the dose-dependent induction of neutrophil migration by binding to lactosylceramide followed by activation of the Src family kinase/PI-3K/G-protein signaling pathway [341]. Furthermore, resolvins are a class of anti-inflammatory lipids derived from omega-3 polyunsaturated fatty acids, and *Candida albicans* can synthesize resolvin E1 that reduces neutrophil chemotaxis in response to CXCL8 but enhances the phagocytosis of Candida [342]. Finally, secretory aspartyl proteinases (Saps) of *Candida albicans* increase in vitro chemotaxis of neutrophils. Sap2 and Sap6 then induce neutrophil migration and the release of chemotactic chemokines [343,344,345,346].

## 12. Effects of Antimicrobial Drugs on Neutrophil Migration

Several studies have described the effects of antimicrobial agents on neutrophil migration, including agents used in the treatment of sepsis:The penicillins carbenicillin and piperacillin together with the carbapenem thienpenem and three cephalosporins (cefotetan, ceftazidime, and moxalactam) had no effect on the migration of neutrophils against *Staphylococcus aureus*. In contrast, the third-generation cephalosporin cefoperazone inhibited neutrophil migration [347];Fusidic acid, rifampicin, and doxycycline can also decrease neutrophil chemotaxis, whereas no inhibition of neutrophil chemotaxis could be detected for penicillins, cephalosporins, nalidixic acid, sulfamethoxazole, or trimethoprim [348];Another study showed that tetracycline, minocycline, and erythromycin could inhibit neutrophil chemotaxis [349];In total, 14 cephalosporins, 11 penicillins, and 1 monobactam were evaluated for their in vitro modulation of murine neutrophil migration [350]. The beta-lactam antibiotics could be classified into distinct groups based on their effects on formylated peptide-directed migration: (i) cephalosporin C and cephaloridine had no effect; (ii) decreased migration was observed for cloxacillin, cefotaxime, ceftazadime, cefuroxime, cephalothin, cephapirin, cephadine, cefoperazone, cefoxitin, ceftriaxone, cefadroxil, cefazolin, penicillin G, methicillin, 6-amino-penicillanic acid, nafcillin, piperacillin, ticarcillin, ampicillin, oxacillin, and aztreonam; and (iii) increased migration was observed for cefsulodin [350];Amoxicillin can increase the neutrophil expression of certain adhesion molecules [351]. However, another study described increased neutrophil migration through endothelial cell monolayers by co-amoxiclav, and this was probably due to effects on both cell types [352];The triazoles itraconazole and fluconazole differ in their effect on chemotaxis; only intraconazole inhibits neutrophil chemotaxis [353,354];With regard to formylated peptide-initiated chemotaxis, amphotericin B mediated inhibition whereas ketoconazole caused enhancement and 5-flucytosine, fluconazole, and ciloflucin had no effects [355];Fluconazole (TLR9), voriconazole (TLR2/4/9), liposomal amphotericin B (TLR4), and caspofungin (TLR2/4/9, dectin-1) increased the expression of various pattern-recognizing neutrophil receptors [356].

We emphasize that there are relatively few studies, hence the effects have mainly been observed in experimental models, the molecular mechanisms have not been characterized, and the possible importance of these effects in sepsis has not been investigated.

## 13. Neutrophil Migration in Certain Groups of Patients with an Increased Risk of Infection

### 13.1. Elderly Patients

Neutrophils from elderly individuals were investigated in several studies (Table 4) [357,358,359,360,361,362,363,364,365,366,367,368,369,370,371,372,373,374,375,376,377]. Functional studies have confirmed that both endothelial adhesion and chemotaxis are altered in the elderly [357,358,359,368,369,370,371]. Elderly individuals show preserved chemokinesis (i.e., random migration) whereas chemotaxis is reduced, and this chemotactic defect can be detected from the sixth decade of life [359]. A possible overall effect is therefore less accurate neutrophil migration towards inflammatory sites [359,369,371]. Increased basal PI3K activity seems to be at least partly responsible for the reduced chemotaxis, and inhibition of PI3Kγ/PI3Kδ was suggested as a possible strategy to restore neutrophil migration [359].

**Table 4 cells-12-01003-t004:** Important aging-associated direct and indirect effects on neutrophil migration.

*Modulation of cell surface molecules, release of soluble mediators*
CD11a and CD11b integrin expression are not altered in elderly, but CD16 expression is significantly reduced [357,358]. Plasma levels of neutrophil elastase are increased, suggesting a preactivated basal state of neutrophils [359]
*Altered effects of receptor ligation or altered downstream signaling*
Fc receptors and formylated peptide receptors show altered downstream signaling through MAP kinases, the Janus kinase (JAK)/Signal transducer and activation of transcription (STAT) and PI3K-Akt pathways [360,361]. These effects are due to altered membrane structure with altered receptor recruitment (including TLRs) to lipid rafts [362,363,364]. The altered signaling possibly influences cytokine/chemokine expression/release [366]. Neutrophils show increased basal PI3K signaling [359] but decreased TLR signaling [365].
*Functional effects*
Neutrophils from elderly do not have general defects in endothelial adherence or transendothelial migration [367,368]. Neutrophil chemotaxis in the elderly is impaired due to increased PI3K signaling [359,369]. This impairment is worsened in patients with pneumonia and the worsening is associated with disease severity [359]. Chemotaxis is decreased by surgery; patients recover partly within six weeks and completely within six months [370]. Chemokinesis (random movements) is increased in the elderly whereas chemotaxis is impaired; the final effect is possibly inaccurate neutrophil migration in the direction of infections [359,369,371]. Murine models of aging have shown high reversed transendothelial migration by neutrophils in inflamed tissues; this is due to the desensitization of CXCR1 by CXCL1 releasing cells localized at endothelial cell junctions [372]. These neutrophils then re-enter the circulation and disseminate into the lungs where they cause organ failure [372].
*In vivo observations*
Aged individuals show increased neutrophil levels in airways/bronchoalveolar lavage fluid [373,374,375]. Elderly patients with pneumococcal pneumonia show increased pulmonary infiltration of neutrophils [376]. Some animal models show reduced neutrophil numbers at infection sites in aged mice (*Staphylococcus aureus* and *Pseudomonas aeruginosa*), even though local chemokine levels are high and circulating neutrophils express high CXCR2 levels [359,371,376]. However, other models of inflammation have given different results [377].

Animal studies have shown dysfunctional neutrophil migration to infection sites together with increased reversed migration in aging, and this relocalization contributes to the pathogenesis of organ failure [373,374,375,376,377]. Several molecular mechanisms are involved in this dysfunction, including: (i) reduced integrin expression that influences the endothelial adhesion and transendothelial migration [357,358]; (ii) altered signaling downstream to several receptors, including receptors for chemotactic mediators and signaling initiated by receptors in lipid rafts [359,360,361,362,363,364]; and (iii) altered basal expression/activation of several intracellular pathways [359,365,366,367,368,369,370,371,372,373,374,375,376,377].

### 13.2. Sepsis, Frailty, and Neutrophils

Frailty should be regarded as a manifestation of unhealthy aging and is defined as an aging-related syndrome of physiological decline with marked vulnerability to adverse outcomes [378]. Frail older individuals often present with an increased burden of symptoms (including weakness/fatigue), medical complexity, and reduced tolerance to medical and surgical interventions. Frailty can be measured based on several factors including comorbidity, use of medication, nutritional status, blood tests, or physical/emotional/cognitive function, and multiple tools for screening and assessment of frailty have been developed (e.g., The Clinical Frailty Scale, CFS) [379,380].

Frail elderly people are vulnerable to adverse health outcomes, and frailty has a prognostic accuracy higher than both SIRS and Quick SOFA (qSOFA) criteria among elderly patients admitted to hospital with suspected infection, implying that frailty as a prognostic factor can be used for risk-stratification [381]. Furthermore, preexisting clinical frailty is associated with worse outcomes, including increased mortality for elderly sepsis patients [382], and frailty together with age and disease severity are identified as predictors of long-term mortality in very old intensive care patients with sepsis [383]. Finally, a longer duration of mechanical ventilation in frail sepsis patients who underwent protocol-based weaning was also observed [384].

Frailty can be associated with systemic signs of inflammation, and aging characterized by additional frailty is associated with abnormalities in neutrophil levels and neutrophil migration [385,386]. Some studies describe that frail elderly people, compared to age- and sex-matched controls, show a stable and low-grade inflammation according to C-reactive protein levels (referred to as inflammageing), and this is associated with an expansion of circulating myeloid cells [387,388]. A positive correlation between the frailty score and neutrophil count was described for frail elderly women, whereas a negative correlation was seen for the frailty score and lymphocyte counts [387]. Other studies have also described associations between white blood cell counts and frailty, particularly neutrophil and monocyte counts [389]. Hence, both total neutrophil counts and neutrophil to lymphocyte ratio are increased in frail elderly people [390].

A previous study described that neutrophils from frail elderly individuals showed increased expression of integrin αM/CD11b and HLA-DR. The frequency of CD11b^+^ neutrophils correlated with plasma TNFα levels whereas the frequency of HLA-DR^+^ neutrophils correlated with circulating mitochondrial DNA [391]. Additional experimental studies suggested that these two soluble markers were responsible for the induction of this neutrophil phenotype. Furthermore, the aging-associated reduction in the accuracy of migration is worsened by frailty, and this inaccuracy seems to correlate both with physical and cognitive markers of frailty [392]. Additional regression analysis suggested that the frailty index together with age and Charlson’s comorbidity index were able to predict neutrophil chemotaxis. Finally, the reduced neutrophil chemotaxis showed no association with systemic signs of inflammation (inflammageing), but it could be reversed by selective PI3K inhibitors [392].

Studies of the effects of physical activity/lifestyle on neutrophil migration and levels of circulating neutrophils have given conflicting results [393,394]. Taken together these studies suggest that lifestyle differences do not have any major impact on neutrophil migration and cannot explain the neutrophil dysfunctions in elderly/frail individual.

### 13.3. Neutrophil Migration in Patients with Myelodysplastic Syndrome

Myelodysplastic syndrome (MDS) is a hematological malignancy characterized by the abnormal differentiation of myeloid cells and usually one or more cytopenias in peripheral blood; the patients have dysplastic neutrophils and an increased risk of severe bacterial infections [395,396]. Various defects in neutrophil migration were described in MDS, especially for patients with advanced disease. First, MDS patients show decreased in vitro neutrophil chemotaxis in response to the formylated peptides CXCL1 and CXCL8 [397]. This reduction is associated with altered signaling through the ERK1/2 and PI-3K-Akt pathways [398]. Second, MDS neutrophils show decreased expression of the αMβ2 integrin that is important for transendothelial migration [399,400]. Third, MDS neutrophils show increased levels of miR-34a and miR-155 that reduce neutrophil migration towards formylated peptides and CXCL8 [401]. Finally, reduced neutrophil migration was also described for the chronic myelomonocytic leukemia variant [402]. Thus, reduced neutrophil migration seems to contribute to the increased risk of severe bacterial infections in these patients.

### 13.4. Neutrophil Migration after Stem Cell Transplantation

Autologous or allogeneic hematopoietic stem cell transplantation is used in the treatment of several hematological malignancies [403]. The increased risk of severe bacterial infections is associated with the duration and severity of the early posttransplant neutropenia [404]. However, reduced neutrophil chemotaxis [405,406] together with reduced CD16 expression [406] can be observed even after neutrophil reconstitution, whereas β2 integrin expression is normal [406]. Other neutrophil effector functions can also be altered in these patients [407,408,409], although the phagocytic capacity seems to be normal [405,408,409]. Finally, neutrophil functions are further modulated during sepsis in these patients [410]. To conclude, reduced neutrophil migration is observed after stem cell transplantation, but it should be emphasized that the available studies are few and the question of patient heterogeneity was not addressed.

## 14. Is There an Effect of Gender with Regard to the Severity and Mortality of Sepsis?

Previous studies suggest that sex hormones are important metabolic regulators in patients with sepsis [411], and several studies have even described gender differences with regard to the severity and/or mortality of sepsis patients [412,413,414,415]. Increased male mortality was described for patients with surgical septic shock [416], and male gender was also described as an independent risk factor for developing post-traumatic sepsis [417] as well as other major infections after surgery [418]. The molecular and cellular mechanisms behind this gender difference were not characterized in detail, but sex hormones are also involved in immunoregulation [419,420] and one hypothesis has therefore been that females develop stronger innate and adaptive immune responses than males. This hypothesis is supported by observations suggesting that estrogens, progesterone, testosterone, and other androgens are involved in the regulation of neutrophil activation and migration [419,420,421,422,423,424]. The targeting of sex hormone metabolism has even been suggested as a possible therapeutic strategy in sepsis [413,415].

## 15. The Metabolic Heterogeneity of Sepsis Patients: A Subset of Patients Has a Systemic Metabolic Profile Similar to SIRS Patients

The systemic metabolic profiles at the time of hospital admission for patients with bacterial infections and either SIRS or sepsis according to the 2016 criteria [1,2] have recently been characterized [425,426]. These two patient groups differed with regard to the amino acid, lysophosphatidylcholine, and sphingolipid metabolism, but a minor subset of exceptional sepsis patients had a metabolomic profile similar to SIRS patients. Sphingolipids (see Section 2.2, Section 3.6, Section 11.3 and Section 11.4), lysophosphatidylcholines (see Section 3.6), and amino acids [427,428,429,430] are important regulators of neutrophil migration, and the metabolic heterogeneity of sepsis patients may therefore reflect heterogeneity with regard to the regulation of neutrophil migration.

## 16. Therapeutic Targeting of Neutrophils in Sepsis: Tried and Suggested Strategies

Several strategies for the inhibition of molecular mechanisms involved in the regulation of neutrophil migration were tried or considered as possible therapeutic approaches in sepsis. Targeting of neutrophil migration as a possible strategy is supported by the observation that reduced chemotaxis is associated with advanced disease and reduced survival in sepsis patients [205,206,431]. Indeed, decreased neutrophil CXCR2 expression in sepsis correlates with the Apache II severity score [219] and there is an association between high levels of circulating OLFM4^+^ neutrophils and mortality [301]. The strategies listed below were considered and/or tried in humans.

*Targeting of lipid mediators.* Targeting of lipids is suggested both by the importance of neutrophil membrane stiffness/lipid profile, neutrophil polarization, lipid rafts, and caveola as well as chemotactic lipid mediators in the regulation of neutrophil migration, and lipids are also important for the endothelial cell stiffness that influence neutrophil transmigration (Section 2.3 and Section 7; [25,26,27,46,47,88,89,97,98,99,100,101,102,103,104,105,106,107,108,109,110,111,112,113,114,148,149,150,151,152,153]). Statins were tried in the treatment of sepsis, especially in patients with septic pulmonary infections [369,432,433]. Simvastatin can improve/increase neutrophil chemotaxis in healthy elderly individuals; this effect is also observed in elderly with less severe sepsis but not in patients with more severe disease. Simvastatin may also improve the prognosis in septic elderly patients with pulmonary infection [433], but additional clinical studies are needed to verify this observation. Finally, the modulation of arachidonic acid metabolism by ibuprofen was also tried in a small study of sepsis patients, but neutrophil migration was not examined for these patients [434].

*Hematopoietic growth factor therapy.* In vivo G-CSF therapy (see Section 2.1 [13,171,172]) does not alter the expression of CD16, CD64, or L-selectin by circulating and bronchoalveolar lavage fluid neutrophils derived from patients with pneumonia [435]. G-CSF therapy of healthy stem cell donors decreases neutrophil chemotaxis and this effect may also contribute to the reduced chemotaxis in sepsis patients even though the G-CSF responsiveness of neutrophils seems to be decreased in sepsis patients compared with healthy individuals [405,436]. Both GM-CSF and the GM-CSF/IL3 fusion protein also decrease neutrophil chemotaxis [437,438]. G-CSF and GM-CSF therapy was tried in sepsis, but a recent met analysis concluded that none of these growth factors should be recommended for the routine treatment of patients with sepsis [439].

*Activated protein C treatment.* Neutrophils express receptors for this coagulation mediator and recombinant human-activated protein C was tried in the treatment of sepsis (see also Section 5); treatment with this agent decreases chemotaxis for neutrophils derived from the circulation and bronchoalveolar lavage fluid in a human endotoxin-induced pulmonary inflammation model [440].

*Polymyxin B hemoperfusion.* This procedure can reduce endotoxin levels but did not provide a benefit for a group of highly selected sepsis patients; the procedure reduced neutrophil CD11b expression but did not alter chemotaxis [441].

*Inhibition of intracellular signaling.* PI3K contributes to organ failure in sepsis, and inhibition of this pathway may be a possible strategy for targeting neutrophils and inhibiting neutrophil migration (Section 13.1) [217,218,317,442]. Furthermore, observations in an animal model of endotoxin-induced inflammation have shown that p38 inhibition suppresses pulmonary neutrophil infiltration [442]. These observations suggest that pathway inhibitors should be further investigated as a possible therapeutic strategy in sepsis.

*Checkpoint inhibitors.* MDSCs can be increased in sepsis, and these cells show increased expression of the PD-1 ligand (see Section 10.5) [307]. The use of T cell checkpoint inhibitors to restore T cell dysfunctions was investigated in two early clinical studies [443,444]; this strategy may have indirect effects on neutrophil migration and/or alter the effects of local neutrophil-induced recruitment of various T cell subsets to inflammatory sites (Section 6) [125,126,127].

*Targeting of single proinflammatory mediators*. The targeting of TNFα [215], IL1, IFNγ [125,126,127,136,137], proteases (Section 5 [115,116,117,118,119,120,121,122,123]), and female sex hormones [413,415,445] has been tried and/or suggested by animal studies, but the results so far have not been convincing, even though IL1 inhibition and immunostimulation seemed to improve survival for subsets of sepsis patients in two of these studies [446,447,448,449,450].

None of these strategies should be regarded as an established treatment for the routine handling of patients with sepsis, but several of them show that such strategies have the capacity to alter neutrophil functionand several of them even show the capacity for neutrophil migration. Furthermore, some of these studies suggest that certain strategies are effective only for subsets of patients. In our opinion, future clinical studies of such therapeutic approaches should therefore have a focus on the biological and clinical heterogeneity of the patients.

## 17. Conclusions

Neutrophil migration is regulated by complex molecular mechanisms involving both endothelial cell adhesion/transmigration and extravascular chemotactic migration. These mechanisms are modulated in sepsis; common observations include reduced neutrophil migration toward infection sites and even reversed migration that possibly contributes to organ failures. However, sepsis patients should be regarded as heterogeneous with regard to neutrophil migration because the migration is possibly also influenced by the infecting agent, antimicrobial treatment, patient age/frailty, and other diseases predisposing to bacterial infections. Future clinical studies should clarify whether the treatment of sepsis patients needs to be individualized based on this heterogeneity.

## Figures and Tables

**Figure 1 cells-12-01003-f001:**
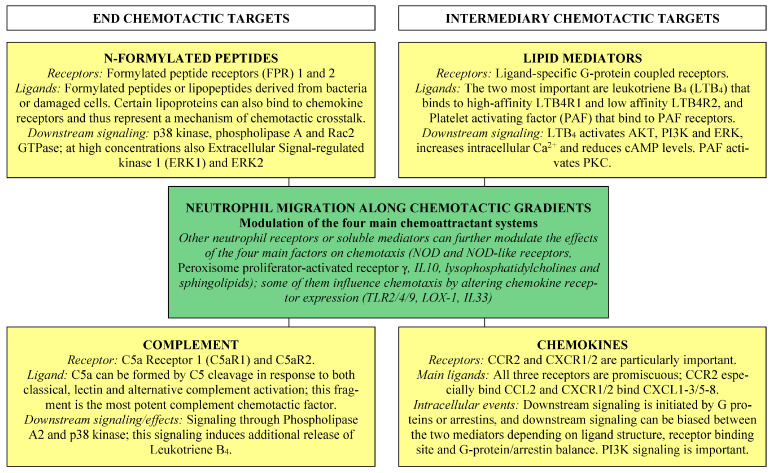
An overview of the classification and important characteristics of the four (see the yellow boxes) main neutrophil chemoattractants (for a detailed discussion with references see Section 3).

**Table 1 cells-12-01003-t001:** Soluble mediators released by neutrophils [125,126,127,128,129,130,131,132,133,134,135]. For the TNF superfamily and angioregulatory/fibroid and other cytokines groups we have indicated important functions in parentheses. Mediators studied mainly in animal models are marked *, and mediators where controversial data exist are marked with (?). Abbreviations: IFN, interferon, and TGF, transforming growth factor.

*Proinflammatory cytokines including interleukins (ILs) and chemokines*
IL1α, IL1β, IL6, IL7, IL9 (?), IL16(?), IL17(?), IL18, Macrophage migration inhibitory factor/CD74-ligand CCL2-4, CCL5 *, CCL 7 *, CCL9, CCL12 *, CCL17-20, CCL22 *, CXCL1-6, CXCL8/IL8, CXCL9-13, CXCL16
*Other immunoregulatory cytokines including interleukins, interferons, and anti-inflammatory cytokines*
IL12, IL21, IL23, IL27, Interferon (IFN)α (?), IFNβ, IFNγ(?) Anti-inflammatory mediators: IL1RA, IL4(?), Transforming growth factor (TGF)β1, TGFβ2
*Hematopoietic growth factors (regulators of myelopoiesis)*
Granulocyte colony-stimulating factor (G-CSF), Granulocyte-macrophage (GM)-CSF(?), IL3(?), Stem cell factor
*The TNF superfamily*
TNFα (proinflammatory) Fas-ligand, TNF-related apoptosis-inducing ligand (TRAIL) (both involved in apoptosis regulation), CD153/CD30 ligand (lymphoid cell regulation), CD40 ligand (T cell regulation) Tumor necrosis factor superfamily member 14 (T cell regulation), Lymphotoxin β (proinflammatory), Receptor activator of nuclear factor kappa-B ligand (RANKL; proinflammatory), APRIL/a proliferation-inducing ligand (regulator of inflammation), BAFF/B cell activating factor (B cell regulators)
*Angioregulatory and fibrogenic cytokines*
Vascular endothelial growth factor (VEGF), Fibroblast growth factor 2 (FGF2), Hepatocyte growth factor (HGF), Angiopoietin 1, TGFα, (angioregulatory), BV8/prokineticin (proinflammatory, chemotactic), Heparin-binding EGF-like growth factor (HBEGF, fibrogenic)
*Other cytokines*
Nerve growth factor (NGF), Brain-derived growth factor (BDNF), Neurotrophin 4 (NT4), Pre-B cell colony enhancing factor (PBEF), Amphiregulin, Midkine, Oncostatin M, Activin A, endothelin

**Table 2 cells-12-01003-t002:** The modulation of neutrophil receptors during sepsis; a summary of important effects on neutrophil migration.

*Altered function of the four main chemotactic mechanisms*	
C5aR	The chemotactic response to Complement 5a receptor is defective in sepsis [170,207,208,209].
CXCR1/CXCR2	Neutrophil CXCR1 surface expression after ligation-induced internalization is quickly restored during sepsis; CXCR1 levels are therefore normal whereas the CXCR2 cell surface expression is decreased due to slower restoration [170,219,220,235,236,237]. This CXCR2 downregulation depends on the activation of TLR2/TLR4/TLR9 [74,79,238] as well as the presence of nitric oxide and TNFα [207]. Other signaling pathways also contribute to CXCR2 downregulation, e.g., soluble guanylate cyclase and PI-3Kγ [92,217,239,240,241].
FPR LTB_4_ receptor	The chemotactic responses to these mediators seem to be reduced in sepsis patients [170].
*Modulation of the four main chemotactic mechanisms*	
TLR2	TLR2 is important especially in gram-positive infections as a receptor for lipoteichoic acid [242]; this receptor reduces neutrophil surface expression of CXCR2 (see above) [196,235,236,237,238,239,240,241].
TLR4	TLR4 is important especially in gram-negative infections as a receptor for lipopolysaccharide [243]; this receptor reduces neutrophil surface expression of CXCR2 (see above) [196,235,236,237,238,239,240,241].
TLR9	This intracellular receptor is important for local recruitment of various immunocompetent cells including neutrophils [196]. TLR9 activation in neutrophils seems to impair chemotaxis and thereby reduce survival for patients with sepsis [79,244].
IL10 receptor	This receptor mediates the inhibition of neutrophil chemotaxis as well as neutrophil cytokine release [91,245], but it can also reduce TLR-induced effects in neutrophils [246,247].
IL33 receptor	IL33R ligation blocks TLR4-mediated CXCR2 internalization [92,93,94,95,96]; neutrophil chemotaxis may thereby be enhanced.
PAFR	Platelet-activating factor (PAF) receptor ligation seems to contribute to the failure of neutrophil migration, and receptor blocking increases survival in animal sepsis models [227,228].
LOX-1	The lectin-like oxidized low-density lipoprotein receptor-1 (LOX-1) is a multiligand receptor [248,249] expressed in neutrophils after TLR2/4 ligation (TLR4 has the strongest effect), it activates NFκB and increases the release of TNFα and IL6 [86,87], but decreases CXCR2 expression [88,89].
PPARγ	Ligation of the nuclear PPARγ inhibits neutrophil chemotaxis and downregulates the expression of several proinflammatory transcription factors (including NFκB and STAT6); its neutrophil expression is increased in sepsis [90,250,251].
PAR	Protease activated receptors (PARs) are expressed by neutrophils, and PAR2 ligation reduces neutrophil migration [121]. Neutrophil expression of PAR2 is increased during sepsis [118,121,122].

**Table 3 cells-12-01003-t003:** Individual molecular markers used to characterize neutrophil subsets and their involvement in neutrophil migration.

Mediator	Expression and Function	**References**
Integrins	αMβ2/Mac-1, α4β2/LFA-1, and α4β2 integrins; important in adhesion/extravasation.	[22,265]
CD54/ICAM-1	A marker of tissue-experienced neutrophils undergoing retrograde migration.	[22]
CD15	This selectin ligand mediates neutrophil adherence to platelets and endothelium.	[266]
CD62L	L-selectin is important for neutrophil–endothelial adhesion and transmigration.	[22]
CD66b	This adhesion molecule and αMβ2 integrins are upregulated after endothelial contact.	[267]
TLR2/9	Receptor ligation inhibits neutrophil chemotaxis through CXCR downregulation.	[234,235,236,237,238,239,240,241,242,243,244]
TLR4	Ligation modulates neutrophil polarization/chemotaxis, possibly byp38 signaling.	[78]
CD14	CD14 is important for TLR4 ligation/signaling.	[268]
TLR7	Animal studies suggest that TLR7 ligation increases neutrophil migration.	[269]
CXCR4	CXCR4/CXCL12 is important especially for retaining neutrophils to the bone marrow.	[5,12,13]
CD16	*Fc receptor CD16*/FcγRIIIa ligation activates neutrophil effector functions; CD16^high^ CD62L^dim^ neutrophils constitute a separate subset.	[270,271,272]
CD64	The CD64 Fcγ receptor show up to 10-fold increased expression during sepsis.	[273]
CD177 and Proteinase 3	CD177^+^ neutrophils constitute 45–65% of circulating neutrophils in most healthy individuals. This cell surface glycoprotein facilitates neutrophil transmigration and tissue invasion together with the coexpressed proteinase 3 (PR3) serine protease.	[270,274]
Olfactomedin 4 (OLFM4)	Olfactomedin mRNA is maximal in immature myelocytes/metamyelocytes. The protein is exclusively expressed in neutrophil granules; animal studies suggest a role in regulation of apoptosis and in regulation of chemotaxis through cathepsin G activation.	[275,276,277,278]
CD10	This cell surface metalloprotease regulates neutrophil adhesion molecule expression.	[279]
CD33	This glycoprotein receptor seems to mediate anti-inflammatory effects.	[280]
Myeloperoxidase (MPO)	MPO^high^ and MPO^low^ subsets are seen after activation and are important for antibacterial effects.	[281]

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
