# Peer review of "The Regulation of Neutrophil Migration in Patients with Sepsis: The Complexity of the Molecular Mechanisms and Their Modulation in Sepsis and the Heterogeneity of Sepsis Patients"

_cells, 2023, doi:10.3390/cells12071003_

Round 1

Reviewer 1 Report

Thank you for submitting this comprehensive review article. It is clearly written and represents a large body of work. 

The review is 36 pages and ~26,000 words long. It reads as a thesis introduction rather than a concise review article. I would typically expect a review article to be max 12,000 words.

The figures are simplistic and there are too many tables. Please consider using inkscape/biorender or similar to generate engaging figures which support the text.

The core objectives of the review are not clear in the introduction

Changes in neutrophil function during sepsis are not discussed until page 18 despite this being the core message of the review

I suggest the following sections can be removed:

- neutrophil heterogenity- this section doesn't clearly link with the rest of the review

- myelodysplastic syndromes/stem cell transplant

- sex hormones: this section contains lots of speculation but very little actual data

In summary I think this review has potential to be an excellent contribution to the field, but is too long, lacks engaging figures and is unfocused in areas.

Author Response

Please see the enclosed cover letter, especially the general comment to Editors and Reviewers on the overall length and our response to specific comment made by the reviewer. 

Reviewer 2 Report

Bruserud et al. submitted an extensive review of the existing literature regarding the regulation of neutrophil migration in patients with sepsis. They thoroughly evaluated published studies regarding neutrophil migration and the factors influencing it.

The review is very relevant to the field but it is very extensive and the sentences are long and complex. There is a lot of information that might be a bit too much for any interested reader. The review should be either shortened extensively or split into two different reviews (one focusing on neutrophil migration in general, the other on neutrophils in sepsis). Up until paragraph 9 (with the exception of parts of paragraph 5) there is only information about neutrophil migration in general but not about its regulation in sepsis, this starts from 9 onwards. Paragraph 11 is very nice. Where is the connection of paragraphs 13.3 (activity and lifestyle), 13.4 (myelodysplastic syndrome), and 13.5 (stem cell transplantation) with sepsis?

Specific comments:

- Table 1: abbreviation SIRS should be either mentioned in introduction or explained in the table.

- Table 1 is also not referred to in the main text.

- The layout of Figure 1 should be changed to have a more uniformed one. Currently in some boxes the words are centered, in others they appear to be justified.

- Also, the black lines surrounding the boxes are not found everywhere.

- L.153/154 the authors talk about heparin but they clearly mean heparan sulfate.

- Figure 2: Why is green box written in italics? Same as with Figure 1, black lines surrounding the boxes are not everywhere, maybe green box can be made a little bit bigger so IL33 fits in the same line as others and text should also be centered vertically.

- L. 203/204: intermediary attractants signaling through PI3K à should also be written in the boxes in Figure 2

- L. 225: authors wrote Annexin A1/Annexin A1, maybe they mean Annexin A1/Annexin A4?

- L. 248/254: authors wrote both classical, lectin and alternative pathways (these are three pathways)

- Paragraph 9.9: exosomes have never been mentioned before in the part about general neutrophil migration

- Paragraph 10.3: CD177 or CD117 as mentioned in line 1076

- L. 1086 OLFM4 as abbreviation is introduced but never used in the whole paragraph

- L. 1095 abbreviation SIRS is not introduced

Author Response

(The authors gave the same response as above.)

Round 2

Reviewer 1 Report

Thank you for submitting this revised manuscript. The content is excellent, and this has potential to be a great contribution to the field, however I feel there is too much content for a single manuscript and the figures remain simplistic without adding value to the reader.

Consider these reviews as exemplars of concise and clear reviews of neutrophil function:

Rosales: https://www.frontiersin.org/articles/10.3389/fphys.2018.00113/full

Filippi: https://ashpublications.org/blood/article/133/20/2149/273847/Neutrophil-transendothelial-migration-updates-and  

Kolaczkowska: https://www.nature.com/articles/nri3399

Overall comments:

-          Consider removing detail of neutrophil function/signalling in health and directing readers to a published review, this will ensure you can focus on the core message of alterations in sepsis. Attempting to cover both normal and sepsis makes the article unfocussed and challenging to read

-          There are multiple spelling mistakes throughout- starting line 19

-          Be clear about where the neutrophils are from in each section- are these all-peripheral blood neutrophils?

Major:

Lines 19-36: Include objective of the review in abstract

Line 58: is there any direct evidence that neutrophil dysfunction leads to worse outcome in sepsis?

Section 3.5: Consider restructuring so that the core message is at the start of the section followed by the evidence

Section 9: include the trialled therapies in humans which pertain to areas you discuss- PI3K, GMCSF etc

Section 10: introductory section should be much smaller, consider directing readers to a published review of neutrophil heterogeneity

Minor

Line 19- spelling and grammatical errors repeated throughout

Lines 42-54 could be removed- detailed clinical discussion of sepsis diagnosis is not relevant to the manuscript

Reviewer 2 Report

I'm happy with the changes and the manuscript improved a lot.

Author Response

We are very grateful for this general comment.

Round 3

Reviewer 1 Report

I agree with the authors that an editorial decision on manuscript length is appropriate.

The manuscript comprehensively covers the literature, but in my opinion attempts to cover too much diluting the impact of the review.